# Of Mice and Men: The Effect of Maternal Protein Restriction on Offspring’s Kidney Health. Are Studies on Rodents Applicable to Chronic Kidney Disease Patients? A Narrative Review

**DOI:** 10.3390/nu12061614

**Published:** 2020-05-30

**Authors:** Massimo Torreggiani, Antioco Fois, Claudia D’Alessandro, Marco Colucci, Alejandra Oralia Orozco Guillén, Adamasco Cupisti, Giorgina Barbara Piccoli

**Affiliations:** 1Nephrology and Dialysis, Centre Hospitalier Le Mans, Avenue Roubillard 194, 72000 Le Mans, France; afois@ch-lemans.fr (A.F.); gbpiccoli@yahoo.it (G.B.P.); 2Department of Clinical and Experimental Medicine, University of Pisa, 56126 Pisa, Italy; dalessandroclaudia@gmail.com (C.D.); adamasco.cupisti@med.unipi.it (A.C.); 3Unit of Nephrology and Dialysis, ICS Maugeri S.p.A. SB, Via S. Maugeri 10, 27100 Pavia, Italy; marco.colucci89@gmail.com; 4Department of Nephrology, National Institute of Perinatology “Isidro Espinoza de los Reyes”, Mexico City 11000, Mexico; ale_gaba@hotmail.com; 5Dipartimento di Scienze Cliniche e Biologiche, Università di Torino, 10100 Torino, Italy

**Keywords:** diet, proteins, pregnancy, chronic kidney disease, nutrition

## Abstract

In the almost 30 years that have passed since the postulation of the “Developmental Origins of Health and Disease” theory, it has been clearly demonstrated that a mother’s dietary habits during pregnancy have potential consequences for her offspring that go far beyond in utero development. Protein malnutrition during pregnancy, for instance, can cause severe alterations ranging from intrauterine growth retardation to organ damage and increased susceptibility to hypertension, diabetes mellitus, cardiovascular diseases and chronic kidney disease (CKD) later in life both in experimental animals and humans. Conversely, a balanced mild protein restriction in patients affected by CKD has been shown to mitigate the biochemical derangements associated with kidney disease and even slow its progression. The first reports on the management of pregnant CKD women with a moderately protein-restricted plant-based diet appeared in the literature a few years ago. Today, this approach is still being debated, as is the optimal source of protein during gestation in CKD. The aim of this report is to critically review the available literature on the topic, focusing on the similarities and differences between animal and clinical studies.

## 1. Introduction

“Der Mensch ist was er isst” (Man is what he eats) wrote Feuerbach and his observation is often cited to underline how important food is to maintaining health [1]. Moreover, our health is also influenced by what our mothers ate during pregnancy, and maternal diet appears to be a factor in some diseases.

The quantity (under- and over-nutrition) and quality of foods we eat, as well as our intake of vitamins and micronutrients, play roles that are only partially understood.

Lessons from disasters such as the famine that affected the Dutch during the Second World War [2,3] and the one that occurred during China’s Great Leap Forward [4,5] in the 1960s, have clearly shown that brutal deprivation, even when of relatively short duration, is not without long-term consequences on kidney function. Proteinuria, hypertension and cardiovascular and kidney diseases are all reported to increase when food intake is grossly inadequate.

Conversely, overnutrition in pregnancy is associated, in the short term, with an increase in the hypertensive disorders of pregnancy and, in the long term, almost paradoxically with the same diseases we find in an under-nourished population [6,7,8,9].

Placental stress is likely to be the common pathway determining the long-term effect of these two opposite challenges (Figure 1) [10].

Analyzing the effect of the different factors in increasing the long-term risk of kidney diseases is not easy. Human nutrition is extremely complex, and is not limited to a mere series of protein, lipid, carbohydrate and energy counts. While during disasters undernutrition usually consists in protein deprivation, in some diseases, such as anorexia, protein intake is less affected and an energy deficit is more evident.

On the other hand, overnutrition, especially in Western countries, is associated with poor quality food, and the effects of quantity, distribution, quality and contaminants become difficult to assess. Furthermore, energy balance and gene expression are known to depend upon the intake of proteins and fat, and this modulation may have different effects on people of different genetic backgrounds [11,12].

If we are to answer at least some of these questions, we need studies employing animal models, as this will enable us to study the effects of single nutrients or of single contaminants (such as drugs or preservative agents), on a genetically identical population.

With regard to the issue discussed, the aim of the present review is to assess the role maternal protein restriction during pregnancy has in determining the offspring’s future kidney health. The subject is a controversial one, as laboratory studies, mainly on rodents, usually apply a severe protein restriction and result in an increased risk of metabolic and kidney diseases in the offspring, while studies on humans are mainly performed with a moderate protein restriction and generally result in a lower risk of pregnancy complications. Conversely, we lack studies on moderate protein restriction in animals, while studies on very low protein intake in humans are usually set in a context of famine or poor access to food.

For this reason, the review is divided into two parts: the first one regards research using animal models, while the second focuses on studies of humans. This narrative revision will analyze the following issues:-Protein-restricted diets in human pregnancy and their link with chronic kidney disease (CKD);-Rodents:⚬Main experimental designs;⚬Commonly used diet formulations;⚬Models of protein deprivation and their effects on offspring’s kidney health.-Humans:⚬The effect on offspring of moderate protein restriction in pregnant chronic kidney disease (CKD) patients;⚬The relevance of different sources of protein and whether it is scientifically acceptable to apply observations derived from animal studies to humans.

## 2. The Various Aspects of Protein-Restricted Diets in Pregnancy in Humans, and Their Link with CKD

At least for nephrologists, protein restriction is not synonymous with protein deprivation, and is usually considered to be an unintended, or extreme and unbalanced reduction in protein intake; in humans this is mainly due to lack of access, usually for economic reasons, to protein-rich foods, mainly of animal origin. In this regard, diets lacking protein are usually also deficient in other important micro and macronutrients, even though they sometimes supply enough calories.

There is a significant overlap between diets that result from a lack of resources and some vegan and plant-based diets. The deficits associated with such diets are highly variable, poorly known and depend on geographic and social context. Iron and Vitamin B12 deficits are the ones most frequently found. Overall, such diets are associated with unfavorable pregnancy outcomes, increased risk of preterm delivery, small for gestational age babies and impaired fetal growth. All these elements are, in turn, associated with a higher risk of metabolic disorders and cardiovascular and renal diseases in adulthood.

Anorexia represents a “rich” counterpart of restricted and unbalanced diets; intake of nutrients, and related deficits, vary widely: they can be multiple in the case of very restricted, rigid, repetitive diets, or there may not be any in the case of associated compulsory vitamin or micronutrient intake. The effect on offspring’s health is mainly mediated by a higher incidence of preterm delivery [13].

*Hyperemesis gravidarum*, usually limited to the first weeks of gestation, can be considered another form of abrupt food deprivation, often associated with severe electrolyte imbalance. Pre-renal acute kidney injury may negatively influence the mother’s health, but the effect on the fetus is usually minor [14].

Outside of pregnancy, moderate to severe protein restriction, when carefully controlled and supplemented if necessary, is presently recognized as the starting point for the clinical management of CKD [15].

Few experiments, most of them carried out by our multicentric group, have investigated the dietary management of women with severe CKD or relevant proteinuria in pregnancy. The diets were usually moderately restricted (protein intake between 0.6 and 0.8 g/kg/day) and were plant based [16]. Supplementation with vitamins, amino and ketoacids was usually added. The somewhat surprising result of better intrauterine growth in pregnant chronic kidney disease (CKD) patients on such diets poses a series of questions, so far unanswered. Of these, the most important is whether the effect on maternal kidney function and on intrauterine growth is linked to protein restriction or to different protein sources (plant-based versus animal), or if it is at least partially caused by the avoidance of food additives and preservation agents frequently used in Western countries, as was suggested by two recent case studies [17]. The evidence from studies of patients following well-planned plant-based diets in pregnancy is reassuring, but the only clearly proven advantage is a reduction in the risk of large for gestational age babies and, possibly, of gestational diabetes [18,19,20]. It is, however, conceivable that other advantages, if they exist, would only be seen in a high-risk situation, like the one that characterizes CKD from its early stages [21,22,23,24,25,26].

## 3. Animal Models: Protein-Restricted Diets in Pregnant Rodents

There are two main approaches to studying protein-restricted diets in rodents: severe protein restriction, most commonly used as a method to induce fetal growth restriction, and variation of protein sources, rarely used, to mimic the dietary interventions prescribed to women.

The first, classic model, applies a severe protein restriction to the pregnant animals, often starting before mating. Table 1 shows some examples of animal feeding formulas. The “regular diet” is usually casein-based with a protein content of 18–22%. The study diet has a protein content ranging from 4 to 10%.

The aim of these models is usually to produce growth-impaired offspring, that, in keeping with the observations on humans mentioned above, are at high risk for the development of kidney abnormalities, or metabolic diseases, cardiovascular diseases, hypertension and kidney disease in adult life. Genetic background modulates the effects, and, for kidney development and diseases, these are more serious when breeds characterized by spontaneous development of CKD are employed.

The second model instead exploits different protein dietary sources in pregnant rodents to study their effects on the offspring. Soya-derived proteins are the most widely used as substitutes for casein. However, reports on pregnant animals are scant and when different protein sources are tested, the amount of total protein in the diet is similar between groups [27,28].

To date, only one study has explored the effect of diet on pregnancy in rats with CKD [29]. Cahill and colleagues studied a model of hereditary kidney cyst disease in Han:SPRD Cy rats, feeding them from two weeks before mating to the end of the weaning period, with either a soya-based or a casein-based diet. The authors found that the plant-based diet improved renal function in the pups. Moreover, renal inflammation and cell proliferation, oxidative stress and proteinuria were reduced in the offspring [29]. This study does however have several limitations: the kidney disease is a hereditary one and renal function was still in the normal range in both groups; secondly, although the sources differed, both diets contained a “normal” amount of proteins.

It is evident that none of these approaches mimic the moderate protein restriction employed for women with CKD in pregnancy. In this case, protein intake is up to 75% of a normal diet and is supplemented with keto-analogues and essential amino acids [16,30]. Keto-analogues are nitrogen-free analogues of essential amino acids that can be added to human low- and very low-protein diets (0.6 g/kg/day of proteins and 0.3 g/kg/day of proteins, respectively) to limit the risk of protein malnutrition [31].

## 4. Models of Protein Deprivation during Pregnancy in Rodents and Kidney Health in Offspring

The earliest studies on low-protein diets in rodents date back to the 1930s [33]. In the Sixties, Zeman observed smaller kidneys with a reduced number of glomeruli in the offspring of dams fed an LPD [34] and undertook a series of experiments to investigate this phenomenon and distinguish it from the effects of calorie restriction during pregnancy [35]. He also hypothesized that smaller kidneys lead to reduced kidney function later in life [35].

In keeping with this observation, in 1988 Brenner proposed that a lower nephron endowment at birth confers increased salt sensitivity and a higher risk of hypertension during adulthood [36].

In 1993, Barker observed the relationship with gestational undernutrition and diseases that develop later in life and formulated the theory of “Developmental Origins of Health and Disease” [37]. Since then, a large body of evidence has been gathered from studies using animal models.

In the 1990s, Langley-Evans produced a series of studies showing that fetal programming by means of a severely protein-restricted maternal diet leads to hypertension, which could be prevented by a blockade of glucocorticoid production [38] or administration of ACE inhibitors to pups [39]. This increase in blood pressure could be, at least in part, caused by an alteration of the normal angiotensin II receptor ratio. In fact, in the kidney cortex of Wistar rat pups whose dams had been fed a severely protein-restricted diet, AT1 receptors were about 62% higher, while AT2 receptors were about 35% lower, with no change in angiotensin II tissue levels or circulating aldosterone levels [40]. Even when a protein-deprived diet was administered only in the second half of pregnancy, the offspring exhibited a higher expression of AT1 receptors and decreased expression of AT2 receptors in the heart [41]. Increased oxidative stress and inflammatory renal cell infiltration contribute to hypertension programming. Increased renal oxidation markers before hypertension onset has been observed in the offspring of mothers fed a low-protein diet during gestation and the administration of antioxidants or mycophenolate during the prehypertensive window was able to prevent high blood pressure in adulthood [42]. These findings were recently corroborated by experiments in which renin-angiotensin system inhibitors were administered after weaning: losartan eliminated inflammatory infiltration and intrarenal renin-angiotensin system (RAS) activation [43] while transient exposure to enalapril reduced glomerular filtration rate (GFR) and prevented the onset of hypertension [44]. Moreover, a role for renal nerves in sodium reabsorption has been postulated: bilateral renal denervation in the offspring of dams fed a low-protein diet was able to increase the fractional excretion of sodium, thereby mitigating the rise of blood pressure [45].

Experimental data suggest that protein restriction timing during gestation influences the severity of hypertension in offspring: feeding dams with a low-protein diet during discrete time frames (i.e., early, mid or late pregnancy) resulted in hypertensive pups but the highest blood pressure levels were observed in pups whose mothers were given a low-protein diet for the entire duration of gestation [46].

In addition, it was observed that severe protein restriction during pregnancy decreased offspring’s life span [47] and impaired nephrogenesis [48], in accordance with Brenner’s hypothesis [49]. The reason why severe protein restriction in the mother turns into a reduced number of glomeruli at birth (20% to 30% fewer glomeruli than in controls) [50,51] has been explained as an imbalance between actively proliferating cells and apoptosis in the metanephros, in favor of apoptosis [51] and with higher p53 expression [52]. Moreover, protein deprivation from conception to the 5th–6th week of gestation alters gene expression in embryonic kidneys, downregulating the expression of prox-1 and cofilin-1, the genes that are pivotal in the normal development of the lymphatic vessels [53] and cytoskeleton [54], respectively [55]. In addition, experimental studies with severe protein restriction in sheep, demonstrated impaired fetal renal microvascular development [56].

Although kidney damage during embryogenesis leads to hypertension during adulthood, the window for programming can extend to the postnatal period and appears to be susceptible to treatment. It was found that feeding the offspring of female rats kept on a low-protein diet during pregnancy, a low-sodium diet or administrating an ACE inhibitor for a short time after weaning was able to prevent a subsequent rise in blood pressure even after discontinuation of treatment [57]. Moreover, cross-fostering pups generated by dams fed a regular 20% protein diet to mothers fed a severely protein-restricted diet, during lactation, resulted in hypertension even in the absence of fetal programming [58]. Conversely, it has been shown that cross-fostering the offspring of mothers with protein malnutrition in pregnancy to mothers fed a normal protein diet was able to normalize the number of glomeruli and normalize blood pressure to the values found in control male rats [59].

However, renal physiology in rodents is different than in humans, as in rodents kidney maturation continues after birth [60,61,62,63] while in humans it ceases by the 36th week of gestation in at-term infants [64] and there is evidence that nephrogenesis continues until the 40th post-natal day only in preterm neonates in whom, however, nephrons remain abnormal [65].

Protein restriction during pregnancy affects male and female offspring differently [66,67]. For instance, hypertension in pups from dams fed a protein-restricted diet has been shown to be glucocorticoid-dependent in males but not in females. The underlying mechanism seems to be a reduced expression of renal AT2 receptors [68]. In addition, male pups from mothers fed a severely protein-restricted diet showed impaired sexual maturation, prostate growth and reproductive ability [69,70]. Similarly, severe maternal protein restriction has been associated with a reduced ovarian reserve in female offspring [71]. These results support the importance of studying the F3 generation, which is the first one not influenced by the mother’s diet during pregnancy, in order to understand the transgenerational effect of gestational protein restriction on kidney structure and function [72] (Figure 2).

Hypertension is not the only cardiovascular risk factor that has been observed in offspring exposed to severe protein restriction in utero: insulin resistance [73], reduced insulin secretion [74] and insulin signaling deregulation [75], reduced pulmonary compliance and higher tissue elastance [76], altered lipid metabolism [77], fatty liver disease [78], cardiac fibrosis [79] and cardiac oxidative stress [80], muscle fiber and neuromuscular junction changes [81], metabolic syndrome [82], altered fat distribution [83], increased susceptibility to vascular injury [84], and alteration of coagulation factors [85] have all been described.

Severe protein restriction in pregnancy can also influence drug response in the offspring. DuBois reported an impaired response to furosemide in programmed Sprague Dawley female rats due to increased renal organic anion transporter 1, irrespective of body weight, an important notion in drug dosing for physicians [86]. In addition, epithelial sodium channel (ENaC) activity, as well as sodium-potassium-chloride cotransporter (NKCC2) activity in the cortical collecting ducts seem to be caused by severe maternal protein restriction, which potentially affects renal sodium handling. The administration of the ENaC inhibitor benzamil, indirectly showed that basal ENaC activity was higher in pups from protein-deprived dams compared to controls [87]. In addition, maternal protein deprivation resulted in increased NKCC2 abundance in the renal medulla and increased chloride transport in the thick ascending limb of the Henle’s loop in their offspring [88].

Maternal protein restriction has also been shown to increase sodium urinary loss [89] and determine an increment of total body sodium content and extracellular fluid volume expansion in the offspring [90]. In order to explain these apparently contrasting observations, Alwasei and colleagues proposed that increased sodium reabsorption in the kidney of the offspring of dams fed a protein-restricted diet is an adaptation to sodium loss due to prenatal injury and sodium losses also stimulate an appetite for salt and increased food intake that lead to accelerated growth and hypertension [90].

The kidneys of pups exposed to severe protein restriction during gestation also showed impaired calcium handling with a decreased passive reabsorption in the proximal tubule, leading to a reduction in femur trabecular bone mass [91].

Finally, it is worth mentioning that there are a small number of studies that were not able to demonstrate the effects of fetal programming by means of severe protein restriction during pregnancy. According to Jones and colleagues, a programmed lower nephron endowment did not influence the onset of kidney disease in diabetic Wistar rats [92]. Zimanyi and colleagues, instead, did not confirm the onset of hypertension and did not find correlations between total filtration surface area and blood pressure in adulthood in pups from Wistar Kyoto dams fed a severe protein-restricted diet during pregnancy and lactation [93]. These results led the authors to suggest that a lower nephron endowment is not sufficient to induce hypertension in adulthood but that these animals are more susceptible to a “second hit”. In fact, although not different from controls in basal conditions, the infusion of advanced glycation end-products (AGEs) in the offspring increased the expression of profibrotic genes and the accumulation of AGEs in the kidney, suggesting increased susceptibility to the development of diabetic nephropathy [94].

Severe protein restriction has been found to alter placental morphology and function in animal models. Several alterations have been described, including reduced vascularization due to increased proinflammatory cytokine secretion by immune cells [95], reduced weight with smaller junctional zone [96], reduced trophoblast giant cells and trophoblast glycogen cells [97], higher oxygen uptake by placental mitochondria that may reflect the uncoupling of respiration and oxidative phosphorylation [98] and downregulated amino acid transport [99].

## 5. Experience in Human CKD Pregnancies with Moderate Protein Restriction and Modulation of Protein Quality

Human pregnancies, like all mammal pregnancies, are characterized by an increased demand for energy and macro- and micro-nutrients. Accordingly, it is generally suggested that protein intake should be increased to meet metabolic requirements. It is recommended that normal protein intake of 0.7–0.9 g/kg/day should be increased by 1 g/day during the first trimester, then by a further 8 g per day in the second trimester, and 23–29 g in the third [100,101]. The modulation of these indications in pregnant women with CKD is not fully agreed but several reports indicate that, in the presence of CKD, a moderate protein restriction might be useful and safe for mother and offspring. The first study of pregnant CKD patients put on moderately protein-restricted diets was published almost a decade ago [17]. Originating as an attempt to balance the contrasting nutritional needs of advanced CKD and pregnancy, 12 pregnancies in 11 patients with CKD stages 3 to 5 and/or proteinuria (>1 g/day), were managed with a 0.6–0.7 g/kg/day vegetarian protein diet with keto-analogue supplementation at increasing dosage throughout gestation. The authors observed no major side effects of this diet and only one pregnancy, in the context of nephrotic syndrome, was terminated. Ten of the 11 babies born were delivered preterm, two were small for gestational age but after birth, the growth curve of all the babies was normal. None of the mothers started dialysis during pregnancy or in the year after delivery [17]. In a subsequent study, these authors applied the same dietary protocol to 24 pregnancies, with 21 control CKD pregnancies with no dietary restrictions [25]. Notably, the number of small for gestational age babies was significantly lower in the diet group than in controls. A follow-up study of the babies from six months to ten years of age found no socialization or schooling problems and similar rates of hospitalization [25]. Expanding the sample size of both cases and controls confirmed the results [22]. These studies included a large spectrum of renal diseases: diabetic nephropathy, glomerulonephritis, kidney transplant, genetic diseases [17,22,25]. In a small, more homogeneous series of patients affected by biopsy-proven focal segmental glomerulosclerosis, with normal kidney function and proteinuria, mothers on the same diet delivered at-term healthy babies without consequences to babies’ growth [21]. Other case reports confirm this trend [102], demonstrating the difference between a controlled diet compared to the “famine” model and the importance of dietary education in order to improve adherence [103].

## 6. What Are the Best Sources of Protein

The differences in quality and properties of animal- and plant-derived protein are summarized in Table 2. Notably, plant-derived proteins are associated with lower phosphate bioavailability [104,105], are less likely to induce acidosis [106], favorably modulate gut microbiota leading to reduced production of uremic toxins [107,108], and are rich in antioxidants. Conversely, animal-derived proteins supply all the essential amino acids. Overall, there are no indications that a plant-based diet is not appropriate in pregnancy. In fact, according to the United States Academy of Nutrition and Dietetics, “Well-designed vegetarian diets provide adequate nutrient intakes for all stages of the life cycle and can also be useful in the therapeutic management of some chronic diseases” [19].

While data on plant-based protein diets were associated with lower mortality in patients with CKD after adjusting for comorbidities and risk factors [109], vegetarian diets during pregnancy carry a risk of nutritional deficits: vitamins B12 [110,111] and D [112], and iron [113] and zinc [114] deficits have been described but these can easily be corrected by close monitoring and well-planned supplementation. These deficits are also common among non-vegetarian mothers and it has been found that vegetarian patients are more compliant in taking supplements [113]. Plant-based diets do not seem to affect pregnancy-related disorders: the incidence of preeclampsia and preterm delivery has been described as equal or lower [115,116,117,118,119] whereas glycemic control seems to be improved by fiber-rich diets [120]. Although we lack proof that a plant-based diet is more beneficial than an omnivorous one, no significant harm was observed.

Soya is the source of vegetable protein most studied in nephrology. Soybeans are a complete source of amino acids, comparable to meat, for which they can serve as a valuable substitute [121]. In renal patients, soya-based diets have been shown to have positive effects on plasma cholesterol and triglycerides [122,123], serum creatinine and phosphates [122], oxidation markers and endothelial function [124,125], glucose metabolism [126] and proteinuria [127,128,129]. Extensive studies on isoflavones, the polyphenolic group of compounds found in soybeans, have shown that they exhibit anti-inflammatory, anticancer, antioxidant and antimicrobial activity [130]. Among them, genistein and daidzein have a structure similar to estradiol and are classified as phytoestrogens because of their affinity for the estrogen receptor [131]. It is worth mentioning, however, that isoflavone metabolism is mediated by the kidney: patients on dialysis show a longer half-life of genistein and daidzein and dialysis clearance is minimal [132].

In pregnant, non-CKD women, soya-based diets have been found to ameliorate glucose homeostasis, lipid profile and antioxidant reserves [133].

The effect of the diet in humans may, however, depend on genetic background, as happens in rodents. In fact, a Canadian study showed that a plant-based diet during pregnancy was associated with a higher risk of delivering small for gestational age babies among European-origin mothers and increased neonatal birth weight among Canadian mothers of South Asian origin [134].

## 7. What Rodent Models Cannot Show Us about Protein Restriction in Human Pregnancy

Although animal models are a cornerstone of experimental research they are far from perfect [135]. Most importantly, protein restriction in pregnant animal models is extreme, between 50% and 70% of the normal protein diet. This configures a model of protein deprivation rather than protein restriction, which is hardly comparable to studies on humans in which protein intake is reduced by a maximum of 25% below baseline.

Secondly, renal development in rodents extends after birth. This differs from humans in whom kidney development is complete by the 36^th^ week of gestation. Moreover, not all rodent strains are equally susceptible to kidney injury, thus the choice of model is critical [136,137]. In addition, severe protein restriction can affect housekeeping gene expression in the offspring, making it crucial to select the reference gene when analyzing genome transcripts [138]. In rodents, the catch-up growth of offspring with intrauterine growth restriction seems to be crucial for developing disease during adulthood [139,140,141,142]. In this context, catch-up growth may be a “second hit” revealing the increased susceptibility of fetal programming, as the offspring would not be able to cope with an increased nutrient intake [143,144].

Finally, despite the large number of models of kidney injury in pregnancy, there are no murine models for chronic kidney disease focusing on maternal diet.

Protein sources during pregnancy are likely to influence offspring health but few studies have addressed this issue. A favorable effect of soya-based compared to casein-based diets has been described in rats in regard to renal inflammation, oxidative stress and endothelial function [29,145].

For humans, our experience is scant. Processed red meat has been associated with the worst outcomes in terms of kidney disease: the generation R study showed that protein intake during the first trimester of pregnancy correlated with renal function in offspring; however, when assessing the different contributions of protein sources, this association was confirmed for vegetable-derived protein but not for animal-derived protein intake [146].

## 8. Future Directions

Research employing animal models is warranted as it will enable us to determine whether a moderate protein restriction during pregnancy in CKD is beneficial.

Theoretically, pregnancy in chronic kidney disease should be studied in experimental models of glomerulonephritis [147] as well as in the setting of a reduction of the renal mass [148,149].

The design we propose is to evaluate the effects of a plant-based, moderately protein-restricted diet during pregnancy and lactation in Wistar rats undergoing 5/6 nephrectomy before mating (thus with moderate kidney function impairment at the time of conception). Pregnancies would be evaluated in terms of abortions, miscarriages, premature delivery, worsening of kidney function or dam’s death. Offspring would be evaluated in regard to body and kidney weight (and their ratio), renal function, presence of physical abnormalities, ability to thrive and develop normally. If possible, the effect of the diet would also be studied in the second and third generation fed different diets (Figure 3).

## 9. Lessons for Clinical Nephrologists

Nephrologists routinely refer to animal models to better understand the pathophysiology and management of kidney diseases. While this approach is pivotal to safely translating basic scientific observations into clinical practice, in some cases, such as the one being discussed, it may be misleading.

Acknowledging the difference between animal models and human situations is crucial if we are to avoid seeing every type of protein restriction in pregnancy as negative.

A semantic problem exists, since the same terms indicate different interventions: “low-protein diets” in rodents are usually synonymous with highly-restricted diets, bordering on protein deprivation; conversely in humans, a “low-protein diet” identifies a protein restriction of 20–25%, often supplemented with essential amino acids or keto-analogues. In this respect, generalization of the current findings in animal models to human pregnancies is not possible. Even in the absence of solid pre-clinical studies, however, some reports indicate that the management of CKD pregnant patients with a balanced, moderate protein restriction is safe and may decrease gestational complications without increasing the risk of renal disease progression in the mother. Larger human studies, and research using animal models are needed to corroborate these promising results (Table 3).

## 10. Conclusions

Severe protein restriction in pregnancy is detrimental for kidney health, in particular if genetic background already predisposes to CKD; there is full agreement between rodents and humans in this respect. However, in pregnant patients affected by CKD, a moderate protein restriction, with a well-planned plant-based diet can be safe for both mother and offspring, decreasing the incidence of pregnancy complications and producing an expected positive effect on offspring’s kidney health.

## Figures and Tables

**Figure 1 nutrients-12-01614-f001:**
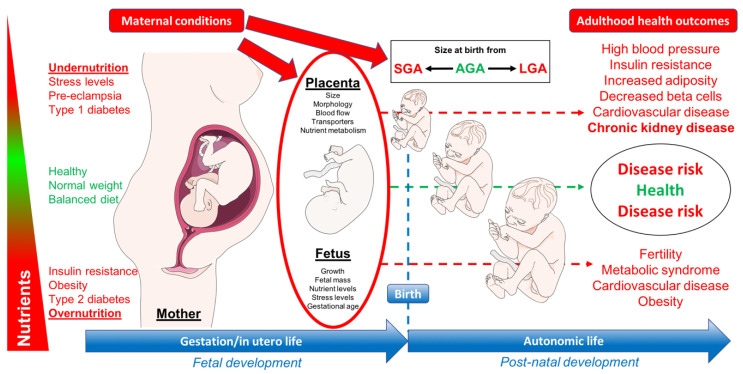
Effect of maternal diet during pregnancy on offspring. Adapted from Chavatte-Palmer P, Tarrade A, Rousseau-Ralliard D: Diet Before and During Pregnancy and Offspring Health: The Importance of Animal Models and What Can Be Learned from Them. *International Journal of Environmental Research and Public Health* 2016, 13. SGA: small for gestational age; AGA: adequate for gestational age; LGA: large for gestational age.

**Figure 2 nutrients-12-01614-f002:**
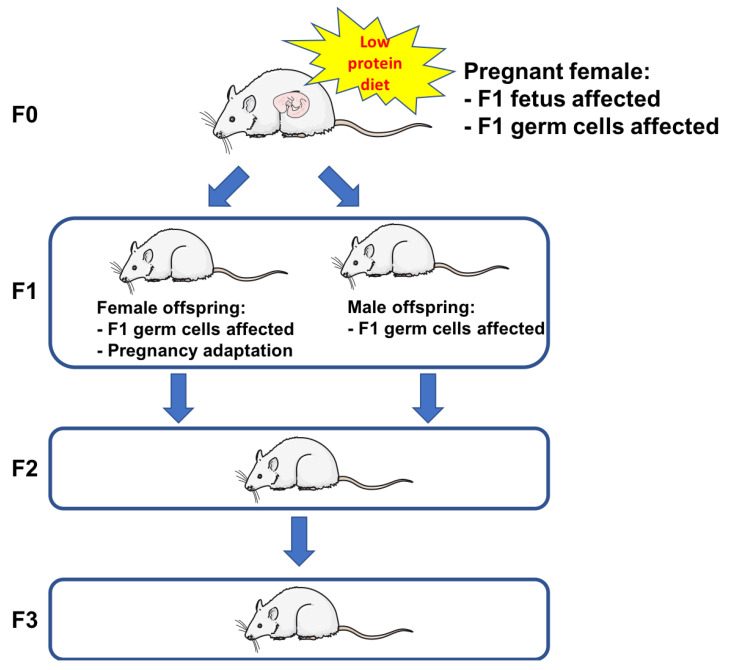
Transgenerational programming. Adapted from Briffa JF, Wlodek ME, Moritz KM. Transgenerational programming of nephron deficits and hypertension. *Seminars in Cell and Developmental Biology*. 2018, Jun 7.

**Figure 3 nutrients-12-01614-f003:**
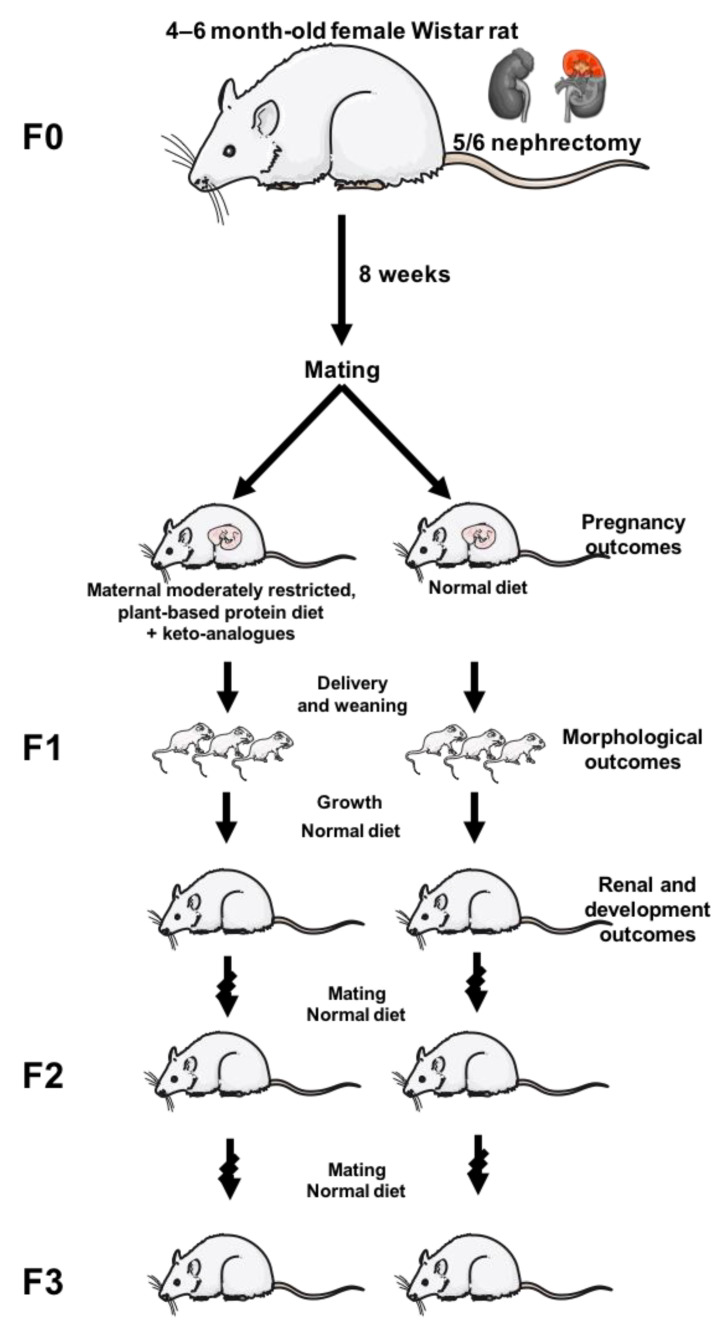
Design of a potential study to test the effectiveness of a moderately reduced-protein, vegetarian, supplemented diet in an animal model of chronic kidney disease.

**Table 1 nutrients-12-01614-t001:** Examples of commercially available normal or low-protein rodent diets.

Manufacturer	Diet	Protein Content (g/100 g)	Fat Content (g/100 g)	Metabolizable Energy (kcal/g)
*Standard Diets*				
LabDiet	5001Laboratory Rodent Diet	24.1	5.0	2.9
Picolab	5053 Rodent Diet 20	20	5.0	3.0
Envigo Teklad	Rodent Diet 8604	24.3	4.7	3.0
Altromin	C1000	20.0	13.0	3,5
American Institute of Nutrition [32]	AIN-93G Rodent Diet for Growth, Pregnancy and Lactation	19.3	16.7	3.8
American Institute of Nutrition [32]	AIN-93M Rodent Diet for Maintenance	14.1	10.0	3.6
*Low-protein Diets*				
Envigo Teklad	TD90016	6.0	5.5	3.8
Altromin	C1003	9.0	13.0	3.6

Data from manufacturers’ websites, accessed on February 10, 2020. LabDiet: https://www.labdiet.com/cs/groups/lolweb/@labdiet/documents/web_content/mdrf/mdi4/~edisp/ducm04_028021.pdf; Picolab: http://www.petfoods.com.mx/PetFoods/LabDiet_ifo_files/5053.pdf; Envigo Teklad: https://www.envigo.com/resources/brochures/rodent-diet-and-ingredient-comparison.pdf; https://www.envigo.com/resources/data-sheets/90016.pdf; Altromin: https://altromin.com/pdf/en/C1000; https://altromin.com/pdf/en/C1003.

**Table 2 nutrients-12-01614-t002:** Main differences between the nutritional profiles of plant- and animal-derived protein patterns.

	Plant-Based Protein Pattern	Animal Protein Pattern
**Energy**	No difference	No difference
**Essential amino acids**	Lacking in methionine and cysteine (lack can be overcome by combining cereals with legumes)	All present
**Fats**	Low, mostly unsaturated	High, mostly saturated
**Fiber**	High	Low
**Iron**	Non-heme iron (reduced bioavailability; Phytic acid and fibers reduce absorption; vitamin C may favor absorption)	Heme iron (high bioavailability)
**Sodium**	Low	High
**Potassium**	High (cooking methods may reduce content)	Low
**Phosphate**	Moderate (in the form of phytic acid, so less easily absorbed)	High
**Production of uremic toxins**	Low	High
**Antioxidants**	High	Low
**Vitamin B_12_**	Low	High
**Calcium**	Low (phytic acid and fibers reduce absorption)	Low, high only in dairy foods
**Folate**	High (cooking methods may reduce content)	Low
**Magnesium**	High	Low
**Zinc**	Low	High

**Table 3 nutrients-12-01614-t003:** Key points for clinical nephrologists.

Rodents	Humans
Severe maternal protein deprivation during pregnancy is detrimental for offspring and leads to an increased risk of cardiovascular and metabolic diseases later in life.	
	A moderate protein restriction in CKD patients has proved to retard the progression of chronic kidney disease and control uremic symptoms [15].
Protein restriction in pregnant animal models is usually severe (50–70%) and unbalanced.	A vegan/vegetarian diet with a 20–25% protein restriction supplemented with keto-analogues in pregnant CKD patients is safe for the mother and the offspring and may help control renal disease without consequences for the newborn [21,22,23,24,25,26].
Consequences of a poor maternal diet on offspring’s health could extend as far as the third generation [72].	
Genetic background and gender modulate the effects of maternal diet on the offspring [66,67,134].	
There are no studies on diet in pregnant rodents with CKD.	
Further research on animal models is needed to better elucidate the mechanisms and long-term consequences on offspring of a moderate protein restriction during pregnancy in CKD in a controlled environment	Longer follow-up studies are needed to study the effects of a moderate maternal protein restriction in the course of CKD pregnancy on offspring’s health in adulthood.

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
