# Peer review of "Of Mice and Men: The Effect of Maternal Protein Restriction on Offspring’s Kidney Health. Are Studies on Rodents Applicable to Chronic Kidney Disease Patients? A Narrative Review"

_nutrients, 2020, doi:10.3390/nu12061614_

Round 1

Reviewer 1 Report

Although there is scientific merit in reviewing the literature regarding understanding the effects of moderate protein restriction during pregnancy (which I appreciate are lacking), the manuscript as written lacks focus, so the reader gets lost as to the subject and point of the review. 

There is lots of miscellaneous information regarding rodent models.  These have little to nothing to do with the point of the review--get rid of it, it detracts--and will lose the interest of any reader. 

A sentence or two recognizing that early expsoures to under or over-nutrition have long term effects on development, growth and health--is fine--and perhaphs even referring to the discovery, Dutch Famine--but you don't need that much--this is already appreciated by scientists--and 'has been appreciated' for awhile.  Your point is that--what the mother is exposed to, results in exposure to her offspring.  The problem now is that women with CKD are treated with mild protein restriction--and we don't know if this will have deterimental effects on offsrping--also there is probably also a need to dissect out milk protein restriction versus expsoure to CKD in utero. 

Organization is needed as content jumps from here to there, and back again.

Authors have sprinkled in flowery writing style here and there--OK if that is your style--but it is used in a way that is detracting, and so greatly decreases the quality of the writing.  Perhaps it is not so much 'flowery' as an overuse of cliches like 'you are what you eat', which is distracting to the reader--adds nothing--and personally detracts from the content, and dece.  The use of flowery writing is appropriate once the content and point are there--cliches are not flowery, but detracting.  

In preparing another draft I suggest removing whole swaths.  Keep in mind much of the rodent stuff that put into this draft has nothing to do with studying effects of diet on offsping.  As a researcher/investigator if I wanted to know about strain differences--this is not the type of paper I would go/search for.  

Decide what your point--objective is, and state it, and then always make sure each section is needed to make your point.

Reviewer 2 Report

In the current review, authors have described the effect of protein restriction in pregnancy and the offspring’s kidney health. The literature is reviewed comprehensively and the presentation is good but the title seems less appropriate and must be more straightforward.

Nephrologists lessons need some more detailed description in the main text to justify presence in the title. Language seems a bit more artistic at some places (for example see line 84).

Round 2

Reviewer 1 Report

The paper should be read by a native English speaker-writter so language and word choice is more consistent with accepted conventions

Author Response

We thank the reviewer for his comments.

            The paper should be read by a native English speaker-writter so language and word choice is more consistent with accepted conventions.

The manuscript has been revised by Susan Finnel.

Susan Finnel is a native English speaker-writer born in the USA. In June 1966, she graduated in English and American literature at the Vassar College, Poughkeepsie, New York. In 1969, she graduated in English and Education at the Long Island University, Brooklyn, New York. She has been our group’s English editor for 5 years.